# Is Countermovement Jump an Indirect Marker of Neuromuscular Mechanism? Relationship with Isometric Knee Extension Test

**DOI:** 10.3390/jfmk9040242

**Published:** 2024-11-18

**Authors:** Esteban Aedo-Muñoz, Jorge Pérez-Contreras, Alejandro Bustamante-Garrido, David Arriagada-Tarifeño, Jorge Cancino-Jiménez, Manuel Retamal-Espinoza, Rodrigo Argothy-Buchelli, Ciro Brito, Pablo Merino-Muñoz

**Affiliations:** 1Escuela de Ciencias de la Actividad Física, El Deporte y la Salud, Facultad de Ciencias Médicas, Universidad de Santiago de Chile, Santiago 8370003, Chile; esteban.aedo@usach.cl (E.A.-M.); ciro.brito@usach.cl (C.B.); 2Escuela de Ciencias del Deporte y Actividad Física, Facultad de Salud, Universidad Santo Tomas, Santiago 8370003, Chile; jperez51@santotomas.cl; 3Escuela de Doctorado de La Universidad de Las Palmas de Gran Canaria (EDULPGC), Las Palmas 35016, Spain; 4Departamento de Educación Física, Deportes y Recreación, Facultad de Artes y Educación Física, Universidad Metropolitana de Ciencias de la Educación, Santiago 7760197, Chile; alejandrobustamanteg@gmail.com; 5Escuela de Kinesiología, Facultad de Ciencias Médicas, Universidad de Santiago de Chile, Santiago 8370003, Chile; david.arriagada@usach.cl (D.A.-T.); jorge.cancino@usach.cl (J.C.-J.);; 6Grupo de Investigación en Rendimiento Físico Militar (Renfimil), Escuela Militar de Cadetes “General José María Córdova”, Bogotá 111211, Colombia; reargothyb@gmail.com; 7Department of Physical Education, Federal University of Juiz de Fora, Governador Valadares 35010-180, Brazil

**Keywords:** biomechanics, kinetic, kinematic, vertical jump, rate of force development

## Abstract

Several studies have shown that force application is influenced by different neuromuscular mechanisms depending on the time of force application analysis in isometric knee extension test (IKE), and a countermovement jump (CMJ) has contributions from knee extension, so some CMJ variables could be indicators of such mechanisms. **Purpose:** The aim of this study was to determine the level of relationship of variables of IKE and bilateral CMJ tests. **Methods**: Male college soccer players (n = 25; corporal mass = 72 ± 8 kg; height = 171 ± 5 cm; age = 22 ± 2 years) performed the IKE at two angles (60° and 75°) on an isokinetic machine and the CMJ on two uniaxial force platforms. To determine the level of relationship, Pearson’s correlation coefficient was analyzed between the test variables. **Results**: Trivial to moderate correlations (r = −0.45 to 0.62; *p* < 0.05) were found between CMJ variables and IKE in both knee angles (60° and 75°); **Conclusions**: The variables of IKE have a trivial to moderate correlation with the variables of CMJ, so the variables of CMJ could not be considered interchangeably with those of IKE and therefore considered indicators of neuromuscular mechanisms isolated from the knee extensor function. Longitudinal design (fatigue or training protocols) should be realized to corroborate these results.

## 1. Introduction

The ability to generate force with the lower limbs is very important in sports actions, such as running, jumping, and changing direction; however, these actions that are performed in various sports or in daily living have a limited amount of time for application of force (50 to 400 ms depending on the technique and phases) [1]. This is why the assessment of force in the specific force-application times of sports or daily actions becomes crucial to evaluate sports performance [2]. To assess muscular strength, both dynamic or isometric and single-joint or multi-joints tests can be performed. Isometric single-joint tests have been used for more than five decades to measure muscle function due to their easy implementation and excellent reliability [3], and it has been evidenced that the application of force or torque over time, quantified as rate of force development (RFD) or rate torque development (RTD) and defined as the first derivative of force or torque over time, respectively, is influenced by different neuromuscular factors [3,4]. RTD and RFD in their early phase (<100 ms) are more influenced by neural mechanisms such as motor recruitment and motor unit discharge rate and in their late phase (>100 ms) by intrinsic muscle properties such as maximal force and muscle thickness [1,4,5]. This has been proven in several studies using isometric knee extension tests (IKE) [1,4,5,6,7]. This influence in different time intervals has been highlighted by several authors [6,7] due to the fact that changes and/or adaptations in different mechanisms could be analyzed without the need for the use of other equipment, such as, for example, the use of electromyography in order to analyze changes in muscle activation [4,8]. However, it cannot be assumed that the same mechanisms have similar influence during dynamic assessments of muscular strength [3,9].

In the dynamic assessments to evaluate the muscular strength of the lower limbs, vertical jumps are commonly used because they do not produce fatigue, are not invasive, and require little time to be applied [10], and countermovement jump (CMJ), which involves a stretch-shortening cycle (eccentric to concentric muscular contraction), is also widely used because it shows a strong relationship with other sport actions, such as acceleration, deceleration, and change of direction [11,12,13]. The jump height (JH), the most used variable from the CMJ and the easiest to calculate, has also been used to assess the neuromuscular state of athletes [14], but sometimes, it is too insensitive for detecting fatigue [15,16], mainly because athletes are able to maintain JH by altering their movement strategies [17]. An advantage offered by the assessment of CMJ using force platforms is that many kinetic variables can be derived from eccentric and concentric phases (sometimes called downward and propulsive phases) [18,19], and these, unlike JH, have been shown to be more sensitive for detecting acute and also chronic changes in the neuromuscular state of athletes, such as fatigue and detraining in specific phases of CMJ [15,20,21]. For this reason, and understanding that CMJ performance has contributions from the knee, some of its variables may have moderate to strong correlations with IKE [22]. Currently, in correlational studies between CMJ variables and IKE, the relationship between JH and the kinetic variables of isometric lower-body single-joint isometric tests on isokinetic machines has been mainly analyzed [22,23,24], and only one study analyzed the relationship between the RTD (i.e., IKE) and RFD (i.e., in CMJ), where it found trivial to small relationships with RTD in isometric single-joint test (hip, knee, and ankle), but it only analyzed the peak RTD and RFD in both tests [25], so the relationship between RTD in time intervals and other variables from CMJ is still unknown.

In the past, for coaches or sport scientists, access to technology such as isokinetic machines and laboratory force platforms has often been limited due to its size and cost; recently, much more economical and ecologically applicable field tools have emerged in sports, such as portable force platforms [26,27], which mainly, due to their low cost and easy transportation, are more accessible to technical teams and sports federations. So, the analysis of the relationship between CMJ variables and isometric tests could deliver information and values for some neuromuscular mechanisms (neural or muscular), delivering information from more applicable assessments within the area of sports biomechanics, physical activity sciences, and/or sports medicine that are more ecological, economical, and practical to perform in the field. From the background, the objective of the present study was to analyze the level of the relationship between different variables of different phases of CMJ and IKE.

## 2. Materials and Methods

### 2.1. Design

The present study was carried out through a quantitative approach with a non-experimental design of a cross-sectional type and correlational scope.

### 2.2. Procedures

The measurements were evaluated through two visits to the laboratory with one week of difference between visits. On the first visit of the laboratory, the informed consent was presented to the participants. Subsequently, familiarization of the experimental tests was carried out: bilateral countermovement jump (CMJ) and the isometric knee extension (IKE) tests, according to methodological recommendations involving RFD and RTD measures [3,28]. On the second visit, the subjects performed a standardized warm-up of 5 min of jogging on an electric treadmill at 8 km/h with no incline, followed by 20 squats and 10 front lunges per profile, followed by 2 min of dynamic lower-limb stretching, followed by a 3 min rest before jump evaluation [29] and later CMJ and IKE in this order. All subjects were informed of the risks, benefits, and objectives (first visit), and they completed an informed consent according to the Helsinki Agreement, which was approved by the local institutional ethics committee (code: 418/2023).

### 2.3. Sample

The sample comprised male college soccer players (n = 25; corporal mass = 72 ± 8 kg; height = 171 ± 5 cm; age = 22 ± 2 years). The players were in a competitive period where they underwent strength and conditioning training on Mondays and technical-tactical training on Tuesdays and Wednesdays and played an official match on Fridays. The inclusion criteria were (i) men with an age range of 18 to 30 years belonging to a university sports team, (ii) playing sports or physical exercise at least 3 times a week, and (iii) not having suffered a lower-limb injury during the last 6 months. The exclusion criterion was presenting any discomfort or pain during the study, either in the hours prior to the study, during warm-up, or during data recording.

### 2.4. Data Recording

#### 2.4.1. Countermovement Jump Recording

Two portable PASPORT force plate platforms were used, namely model PS-2141 (PASCO^®^ Scientific, Roseville, CA, USA), validated for vertical jumps [26,27], with a sample frequency of 1000 Hz with Pasco Capstone software version 2.3.1.1 (PASCO Scientific, Roseville, CA, USA). Their data were exported into a spreadsheet. Subjects were instructed to keep their hands on their hips (Figure 1) throughout the jump to focus only on the force generated by the lower extremities [30] and to jump as fast and as high as possible with their preferred depth [31], and they could choose the amplitude of the countermovement to avoid changes in the coordination pattern of the jump [32]. The subjects performed three attempts and rested for at least 15 s between attempts while data were stored.

#### 2.4.2. Isometric Knee Extension Recording

The isokinetic equipment HUMAC NORM^®^ testing and rehabilitation system (Model 502140, Stoughton, MA, USA) with a sample frequency of 1250 Hz was used for IKE. The positioning was performed according to the HUMAC NORM System User’s Guide provided by the manufacturer. In each evaluation, each participant was positioned for alignment of the joint axis with the mechanical axis of the isokinetic equipment in relation to his anthropometric measurements, ensuring the participants’ comfort and safety and the reliability of data collection. Each participant performed four attempts with a duration of 4 s with their dominant leg (preferred for kicking a soccer ball), with a 30 s pause between attempts and 3 min between angles with a knee angle of 60° and 75° because at these angles, the peak torque and RTD were higher compared to other angles [4]. The neutral approach indication was given to push as fast and hard as possible and place the hands on the equipment as shown (Figure 1).

### 2.5. Data Processing

#### 2.5.1. Countermovement Jump Processing

The spreadsheets were processed through the MATLAB^®^ software by a routine created by the authors (R2021a; The MathWorks, Inc., Natick, MA, USA). To detect the jump’s onset, the method of three standard deviations and a time window of 2 s prior to the jump was used. For the identification of phases (unloading, yielding, braking, and concentric), the method proposed by Harry et al. [33] was used. For the analysis, the following variables were used: peak force, jump time, time for phases, mean force for phases, peak RFD for braking phase, and net impulse for phase (or defined integral). The jump height was calculated using impulse method [34]. For the analysis, the average of three attempts was used.

#### 2.5.2. Isometric Knee Extension Processing

For the IKE, the signals were exported to a spreadsheet and finally processed through the MATLAB software by a routine created by the authors (R2021a; The MathWorks, Inc., Natick, MA, USA). The signal was resampled to 1000 Hz. The signal was filtered by a Butterworth low-pass filter of 4 orders with zero lag and a cut frequency of 20 Hz [35]. The first value at 1 newton was identified as the start of the test. For the analysis, the following variables were used: peak torque, RTD in windows of 50 ms up to 200 ms, and the instantaneous isometric peak RTD (PRTD). The attempt with the highest peak force was used for the analysis. The signal shape and variables are given visually in Figure 2.

### 2.6. Statistical Analysis

The normality of the variables was analyzed using the Shapiro–Wilk test, where the assumption of normality was checked (*p* > 0.05). All descriptive statistics are presented as mean and standard deviation. The relationship between variables was analyzed through Pearson’s correlation coefficient, where values were qualitatively categorized as trivial (0.00–0.09), weak (0.10–0.39), moderate (0.40–0.69), strong (0.70–0.89), and very strong (0.90–1.00) as well as negative values. The alpha was set at 0.05. All statistics were conducted in SPSS software version 25.

## 3. Results

In Table 1 are the descriptive statistics of both tests. In Table 2 are the correlation coefficients between kinematic-related variables of CMJ and IKE. Trivial to moderate correlations were found, and only moderate correlations showed statistical significance (*p* < 0.05).

Table 3 shows the correlation coefficients between the related kinematic variables of CMJ and IKE. Trivial to moderate correlations were found, and only moderate correlations showed statistical significance (*p* < 0.05).

## 4. Discussion

The present study aimed to determine the level of relationship between CMJ and IKE variables. The main findings showed trivial to moderate correlations between the variables of both tests.

Various studies have analyzed the relationship between CMJ and IKE, but most studies only analyze the JH [22,23,24,25], and to our knowledge, only one analyzed the correlation of other variables of CMJ with IKE [25]. Regarding the association between JH and IKE variables, some studies have found similar correlations [22,25] and other higher correlations [23,24]. The latter may have found this due to various methodological factors as well as data analysis. De Ruiter et al. (2006) found a moderate to strong correlation between JH and momentum at 40 ms normalized by time at peak torque, but the knee angle at the jump (90° and 120°) was controlled [24]. Laett et al. (2021) found trivial to moderate correlations (*p* > 0.05) between JH and the RTD accumulated in windows of 50 ms from onset to 250 ms. However, they found moderate correlations when the RTD was normalized by peak torque. On the other hand, they used an individualized angle to perform the isometric test (optimal angle through dynamic test) [23]. These methodological and data analysis differences could explain the differences compared to our results, where a controlled knee angle in the squat and adjusted knee angle in IKE could have increased the correlation between tests.

To our knowledge, only Van Hooren et al. (2022) analyzed the correlations between the kinetic variables of CMJ and IKE, finding trivial to weak correlations that agree with our findings. However, they only analyzed the relationship between PRFD and PRTD. They mentioned that their trivial to moderate correlations could be explained by motor control strategies, where subjects with poor inter-muscular coordination but with a great RTD may not be able to transfer their ability into movements that require greater coordination (dynamic multi-joint) [25]. For example, a recent review of the relationship between isometric and dynamic force showed moderate to very strong correlations between the kinetic variables (peak force and RFD) of isometric multi-joint tests, such as mid-thigh pull and isometric squat, with SJ jump height [36], showing an increase in correlation between tests due to biomechanical similarity (i.e., the same joints are involved in both tests). In the same line, the application of force in the vertical jump depends on three joints (hip, knee, and ankle) [37,38], so the dynamic RFD could have different contributions along with the added variability between subjects, where different joint contributions have been observed between good and bad jumpers (depending on jump height) [38]. In a recent study, the peak torque of hip was the variable with the strongest correlation with the JH during CMJ, corroborating this hypothesis [39].

Another factor that could explain the strength of the correlation is the type of contraction in the single-joint test, where studies have found weak to very strong correlations between dynamic knee extension tests (peak torque and power) and jump height [40,41], the same phenomenon that happens with the sprint, where IKE has a trivial correlation, and fast concentric has a moderate correlation [42]. Another study correlated knee isometric flexion PT and RTD with sprint performance at 30 m and found an explained variance of isometric in sprint between 0 and 28% [43]. That study analyzed the relationships between these variables and PRTD in knee extension and flexion and plantar flexion, finding trivial to weak correlations with JH, JT, TPF, and PRFD [25] and reaffirming that different physiological mechanisms modulate variables derived from dynamic and isometric actions [9], and therefore, monoarticular isometric testing would give poor information on RFD in sport movements [25,42,43,44,45].

One interesting aspect was that most moderate correlations were found between the eccentric variables of jump, and only concentric time had this correlation strength. One study found a similar neuromuscular response after eccentric and isometric exercise compared with concentric, explaining the torque–time integral between protocols [46]. This could explain our result, indicating more similarity between isometric and eccentric contractions than concentric. Also, the muscular strength of knee could be contributing more during eccentric phase and the ankle and hip during concentric phase [47].

One limitation of this research is the study design, and the results should be corroborated with an experimental and longitudinal design; for example, future research could simultaneously analyze both pre- and post-intervention tests (fatigue or training protocol). Other limitations correspond to methodological concerns about preferred depth countermovement in CMJ; because this affects the CMJ variables, it would be necessary to find a standardized methodology for assessment of CMJ. Also, unilateral CMJ should be tested since IKE was unilateral.

## 5. Conclusions

The variables of IKE have a trivial to moderate correlation with the variables of CMJ, so the variables of CMJ could not be considered interchangeably with those of IKE and therefore considered indicators of neuromuscular mechanisms isolated from the knee extensor function. A longitudinal design should be realized to corroborate these results.

## Figures and Tables

**Figure 1 jfmk-09-00242-f001:**
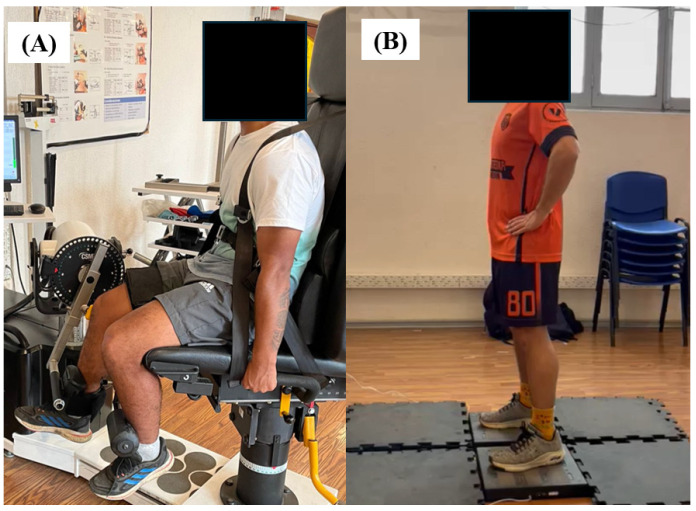
Start position in isometric knee extension (**A**) and countermovement jump (**B**).

**Figure 2 jfmk-09-00242-f002:**
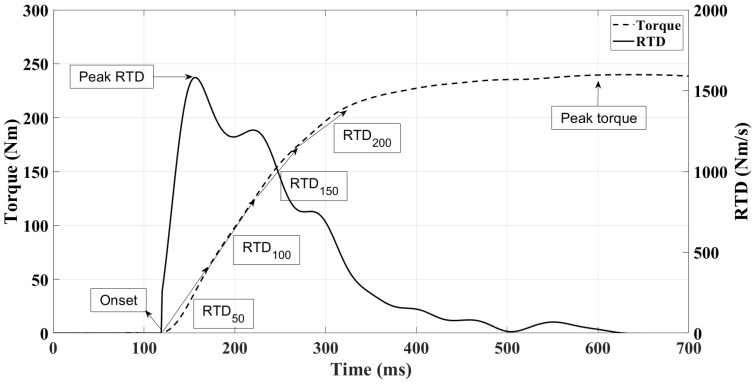
Signal shape and variables of isometric knee extension torque and isometric rate of torque development (RTD).

**Table 1 jfmk-09-00242-t001:** Descriptive statistics of CMJ and IKE test.

Countermovement Jump (CMJ)				Isometric Knee Extension (IKE)			
Variables		Mean	±SD	Variables		Mean	±SD
Kinematic	Jump height (m)	0.284	0.060	60 degrees	PT (N·m)	236	29
	Jump time (s)	0.688	0.076		PRTD (N·m/s)	5998	1546
	Time unloading (s)	0.167	0.039		RTD_50_ (N·m/s)	1119	383
	Time yielding (s)	0.152	0.028		RTD_100_ (N·m/s)	1077	260
	Time braking (s)	0.118	0.028		RTD_150_ (N·m/s)	827	128
	Time concentric (s)	0.250	0.034		RTD_200_ (N·m/s)	580	173
Kinetic	Peak force (N)	1649	228	75 degrees	PT (N·m)	216	33
	RFD unloading (N/s)	−53	13		PRTD (N·m/s)	5449	1682
	RFD yielding (N/s)	61	13		RTD_50_ (N·m/s)	810	478
	RFD braking (N/s)	139	49		RTD_100_ (N·m/s)	947	270
	PRFD (N/s)	8946	3741		RTD_150_ (N·m/s)	811	224
	MF unloading (N)	342	56		RTD_200_ (N·m/s)	571	167
	MF yielding (N)	412	66				
	MF braking (N)	705	132				
	MF concentric (N)	676	127				
	Impulse unloading (N·s)	56	12				
	Impulse yielding (N·s)	62	14				
	Impulse braking (N·s)	82	20				
	Impulse concentric (N·s)	166	21				

PT, peak torque; PRTD, peak rate of torque development; RTD50, rate of torque development 0–50 ms; RTD100, rate of torque development 50–100 ms; RTD150, rate of torque development 100–150 ms; RTD200, rate of torque development 150–200 ms; PRFD, peak rate of force development; RFD, rate of force development; MF, mean force.

**Table 2 jfmk-09-00242-t002:** Correlation matrix between kinematic-related variables of CMJ and IKE at 60° and 75°.

		60° Knee Angle						75° Knee Angle					
Variables		PT	PRTD	RTD_50_	RTD_100_	RTD_150_	RTD_200_	PT	PRTD	RTD_50_	RTD_100_	RTD_150_	RTD_200_
Jump height	r	−0.15	−0.11	0.08	−0.28	−0.33	−0.07	−0.23	−0.13	0.25	−0.22	**−0.40**	−0.29
	*p*	0.47	0.59	0.71	0.17	0.11	0.76	0.28	0.53	0.23	0.30	**0.05**	0.17
Jump time	r	0.14	0.08	−0.12	0.00	**0.44**	**0.46**	0.05	0.03	−0.17	0.04	0.07	**0.46**
	*p*	0.50	0.70	0.57	0.99	**0.03**	**0.02**	0.81	0.91	0.42	0.86	0.74	**0.02**
Time unloading	r	−0.16	−0.07	−0.13	−0.11	0.26	0.00	0.06	−0.01	−0.17	0.05	0.23	0.25
	*p*	0.46	0.75	0.52	0.59	0.21	0.99	0.78	0.98	0.41	0.80	0.27	0.22
Time yielding	r	0.22	0.13	−0.03	0.06	0.33	**0.45**	0.14	0.13	0.07	0.09	−0.04	0.34
	*p*	0.29	0.54	0.87	0.78	0.10	**0.02**	0.50	0.53	0.75	0.69	0.86	0.10
Time braking	r	0.20	−0.01	−0.09	0.00	0.03	0.29	−0.09	−0.04	−0.07	−0.13	−0.14	0.04
	*p*	0.35	0.97	0.68	0.99	0.88	0.16	0.66	0.84	0.76	0.55	0.49	0.84
Time concentric	r	0.15	0.18	0.00	0.05	0.36	**0.41**	0.00	0.01	−0.16	0.02	0.02	**0.41**
	*p*	0.47	0.40	0.99	0.80	0.08	**0.04**	1.00	0.98	0.44	0.94	0.92	**0.04**

Bold values mean significant bilateral *p*-value <0.05; r, Pearson correlation coefficient; *p*-value; PT, peak torque; PRTD, peak isometric rate of torque development; RTD_50_, rate of torque development 0–50 ms; RTD_100_, rate of torque development 50–100 ms; RTD_150_, rate of torque development 100–150 ms; RTD_200_, rate of torque development 150–200 ms.

**Table 3 jfmk-09-00242-t003:** Correlation matrix between force–time-related variables of CMJ and IKE at 60° and 75°.

		60° Knee Angle						75° Knee Angle					
Variables		PT	PRTD	RTD_50_	RTD_100_	RTD_150_	RTD_200_	PT	PRTD	RTD_50_	RTD_100_	RTD_150_	RTD_200_
Peak force	r	0.23	0.14	0.04	**0.41**	0.18	0.09	0.36	0.15	−0.17	0.27	0.50	0.15
	*p*	0.27	0.51	0.85	**0.04**	0.40	0.67	0.07	0.49	0.41	0.19	0.01	0.49
RFD unloading	r	−0.16	−0.20	−0.20	−0.19	0.22	0.08	0.10	−0.15	−0.33	0.14	0.23	0.40
	*p*	0.45	0.34	0.34	0.37	0.29	0.71	0.64	0.48	0.11	0.50	0.27	0.05
RFD yielding	r	−0.22	−0.07	0.03	0.14	−0.39	**−0.45**	0.03	0.08	0.20	0.04	0.06	−0.17
	*p*	0.29	0.73	0.89	0.50	0.06	**0.02**	0.88	0.71	0.35	0.87	0.77	0.43
RFD braking	r	−0.30	−0.09	0.00	0.15	−0.20	**−0.44**	0.01	−0.08	0.03	0.16	0.05	−0.02
	*p*	0.15	0.67	0.99	0.47	0.34	**0.03**	0.96	0.72	0.87	0.44	0.80	0.91
PRFD	r	0.01	0.13	0.18	0.40	−0.11	−0.33	0.26	0.09	−0.11	0.37	**0.39**	0.04
	*p*	0.97	0.55	0.40	0.05	0.60	0.11	0.20	0.68	0.61	0.07	**0.05**	0.84
MF unloading	r	0.37	0.37	0.32	**0.62**	0.07	0.02	**0.41**	0.19	0.00	**0.40**	0.35	0.17
	*p*	0.07	0.07	0.13	**<0.01**	0.74	0.91	**0.04**	0.35	0.99	**0.04**	0.08	0.41
MF yielding	r	0.32	0.39	0.35	**0.54**	0.15	−0.07	**0.41**	0.24	0.07	0.33	**0.44**	0.04
	*p*	0.12	0.06	0.09	**0.01**	0.47	0.73	**0.04**	0.24	0.73	0.11	**0.03**	0.85
MF braking	r	0.06	0.12	0.12	0.33	−0.01	−0.19	0.24	0.09	−0.18	0.24	**0.46**	0.04
	*p*	0.77	0.56	0.56	0.11	0.98	0.38	0.25	0.67	0.38	0.25	**0.02**	0.84
MF concentric	r	0.06	−0.03	0.07	0.06	−0.19	−0.09	0.13	0.10	0.22	0.02	0.02	−0.28
	*p*	0.78	0.88	0.75	0.76	0.37	0.68	0.54	0.65	0.29	0.91	0.94	0.17
Impulse unloading	r	0.07	0.16	0.06	0.30	0.32	0.04	0.36	0.14	−0.18	0.36	**0.51**	**0.40**
	*p*	0.75	0.44	0.79	0.15	0.12	0.86	0.08	0.50	0.39	0.07	**0.01**	**0.04**
Impulse yielding	r	**0.44**	**0.41**	0.27	**0.46**	0.38	0.29	**0.42**	0.27	0.12	0.28	0.29	0.29
	*p*	**0.02**	**0.04**	0.20	**0.02**	0.07	0.16	**0.03**	0.20	0.57	0.17	0.15	0.16
Impulse braking	r	0.24	0.08	0.01	0.23	0.04	0.16	0.04	−0.01	−0.19	0.00	0.17	0.03
	*p*	0.25	0.69	0.98	0.26	0.85	0.45	0.85	0.98	0.36	1.00	0.42	0.89
Impulse concentric	r	0.29	0.17	0.14	0.19	0.09	0.26	0.22	0.17	0.21	0.06	0.06	−0.05
	*p*	0.16	0.41	0.50	0.37	0.66	0.21	0.30	0.42	0.31	0.79	0.79	0.83

Bold values mean significant bilateral *p*-value <0.05; r, Pearson correlation coefficient; *p* value; PT, peak torque; PRTD, peak isometric rate of torque development; RTD_50_, rate of torque development 0–50 ms; RTD_100_, rate of torque development 50–100 ms; RTD_150_, rate of torque development 100–150 ms; RTD_200_, rate of torque development 150–200 ms; PRFD, peak rate of force development; RFD, rate of force development; MF, mean force.

## Data Availability

Data will be made available under reasonable request.

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
