# Peer review of "Is Countermovement Jump an Indirect Marker of Neuromuscular Mechanism? Relationship with Isometric Knee Extension Test"

_jfmk, 2024, doi:10.3390/jfmk9040242_

Round 1
Reviewer 1 Report
Comments and Suggestions for Authors
This is an interesting study investigating the relationship between CMJ variables and IKE variables in soccer players. Although the study is well designed, there are some major concerns regarding the methodology used. Bellow Authors may find my comments.
Major concerns:
1. From the introduction is not clear why it is important to investigate the correlation between an isometric single joint movement (IKE) with a multi joint slow stretch shortering cycle action (CMJ). The hypothesis of the study is not strong and needs deeper analysis into the duration of both movements, the connection with soccer performance and the function of the neural system during IKE and CMJ.
2. From the methods is clear that only the best IKE was used in the analysis but it is not clear whether the best CMJ JH was used in the analysis. This is a cross sectional study. It is like taking a photograph of the athletes and try to make results. Authors should considering changing the approach in data analysis and use all efforts (both in IKE and CMJ) as a mean value in an attempt to present the current neuromuscular state of the athletes (https://doi.org/10.1016/j.jsams.2016.08.011, https://doi.org/
10.3390/sports12030079).
3. In line with my previous comment, unless I missed it inside the text, what was the training period of the athletes? The training period may significantly affect RFD or RTD of the players or even the CMJ performance.
4. Correlation analysis is good for investigating the relationships between variables. But, regression analysis may provide more useful insights in this study. Have Authors tried regression analysis between variables from IKE and CMJ? This might strengthens the results of the study.
Lines 57-61: Please, re write the sentence here, it is not clear the message that authors are trying to send.
Line 88: Keep the abbreviations inside the manuscript.
Line 99-100: Please, re write the sentence.
Lines 123-126: It is not clear what actually happened in CMJ evaluation Please, re write the sentence. Also, what jump was used for the analysis?
Line 131: Change the way you start the paragraph.
Line 138: How authors found the dominant leg?
Lines 139-140: This statement needs a reference.
Author Response
Comments 1: From the introduction is not clear why it is important to investigate the correlation between an isometric single joint movement (IKE) with a multi joint slow stretch shortering cycle action (CMJ). The hypothesis of the study is not strong and needs deeper analysis into the duration of both movements, the connection with soccer performance and the function of the neural system during IKE and CMJ.
Response 1: Information is added in the introduction and re write various phrases.
Comments 2: From the methods is clear that only the best IKE was used in the analysis but it is not clear whether the best CMJ JH was used in the analysis. This is a cross sectional study. It is like taking a photograph of the athletes and try to make results. Authors should considering changing the approach in data analysis and use all efforts (both in IKE and CMJ) as a mean value in an attempt to present the current neuromuscular state of the athletes (https://doi.org/10.1016/j.jsams.2016.08.011, https://doi.org/
10.3390/sports12030079).
Response 2: The mean value was used in CMJ according to recommendations and best attempt (peak force) was used based in other studies analyzing RTD (Cossich V, Maffiuletti NA. Early vs. late rate of torque development: Relation with maximal strength and influencing factors. J Electromyogr Kinesiol 2020; 55: 102486.)
Comments 3: In line with my previous comment, unless I missed it inside the text, what was the training period of the athletes? The training period may significantly affect RFD or RTD of the players or even the CMJ performance.
Response 3: The training period and training courses are added in the week in sample section.
Comments 4: Correlation analysis is good for investigating the relationships between variables. But, regression analysis may provide more useful insights in this study. Have Authors tried regression analysis between variables from IKE and CMJ? This might strengthens the results of the study.
Response 4: Since regression analysis is for prediction, and correlation only reach moderate levels, coefficient of determination would be weak, thinking that is squared correlation coefficient.
Comments 5: Lines 57-61: Please, re write the sentence here, it is not clear the message that authors are trying to send.
Response 5: Was rewrite
Comments 6: Line 88: Keep the abbreviations inside the manuscript.
Response 6: Was corrected
Comments 7: Line 99-100: Please, re write the sentence.
Response 7: Was re write
Comments 8: Lines 123-126: It is not clear what actually happened in CMJ evaluation Please, re write the sentence. Also, what jump was used for the analysis?
Response 8: Was re write and was used the average (in data processing section).
Comments 9: Line 131: Change the way you start the paragraph.
Response 9: Was changed
Comments 10: Line 138: How authors found the dominant leg?
Response 10: Was added
Comments 11: Lines 139-140: This statement needs a reference.
Response 11: Was added a reference
Reviewer 2 Report
Comments and Suggestions for Authors
I thank the editor for giving me the opportunity to review this work. I hope my contribution will be helpful.
Abstract:
You immediately start discussing the objective without explaining the reasons how there could be elements in these two tests that might have correlations, and why it might be worth studying this correlation.
Lines 43-45:
Here I would suggest including some examples of tests.
General considerations:
As you also mentioned in the discussion, there are many differences in the methodology for assessing the bilateral countermovement jump. This perhaps represents the biggest limitation. It would be necessary to find a standardized methodology for assessment when correlating this data with other variables.
The rationale behind the study does not seem very clear. It is not intuitive to think that CMJ and IKE (as they were measured) are two values that could have a correlation. Even if a significant correlation had been found, it would be reasonable to think it might be a spurious correlation (illusory correlation). It would be better to expand the section explaining why you believe these variables could be correlated.
Author Response
Comments 1: You immediately start discussing the objective without explaining the reasons how there could be elements in these two tests that might have correlations, and why it might be worth studying this correlation.
Response 1: Is added a background in the abstract
Comments 2: Lines 43-45: Here I would suggest including some examples of tests.
Response 2: Was added
Comments 3: As you also mentioned in the discussion, there are many differences in the methodology for assessing the bilateral countermovement jump. This perhaps represents the biggest limitation. It would be necessary to find a standardized methodology for assessment when correlating this data with other variables.
Response 3: Was added a limitation.
Comments 4: The rationale behind the study does not seem very clear. It is not intuitive to think that CMJ and IKE (as they were measured) are two values that could have a correlation. Even if a significant correlation had been found, it would be reasonable to think it might be a spurious correlation (illusory correlation). It would be better to expand the section explaining why you believe these variables could be correlated.
Response 4: Was added in introduction
Round 2
Reviewer 1 Report
Comments and Suggestions for Authors
no comments
Reviewer 2 Report
Comments and Suggestions for Authors
After the corrections, the work seems more readable and the rationale behind the study seems more understandable.
Line 25-26: Now it is more clear the reason why you wanted to investigate the correlation.
Line 54-55: The added articles strengthen the reasoning
Line 257-258: This is the biggest limitation and it is good to specify it.